# Murine liver repair via transient activation of regenerative pathways in hepatocytes using lipid nanoparticle-complexed nucleoside-modified mRNA

Fatima Rizvi [1,4], Elissa Everton [1,4], Anna R. Smith [1], Hua Liu[1], Elizabeth Osota[1], Mitchell Beattie[2], Ying Tam[2], Norbert Pardi[3], Drew Weissman[3] & Valerie Gouon-Evans [1✉]

Induction of intrinsic liver regeneration is an unmet need that can be achieved by temporally activating key hepatocyte regenerative pathways. Here, we establish an efficient, safe, non-integrative method to transiently express hepatocyte-growth-factor (HGF) and epidermal-growth-factor (EGF) in hepatocytes via nucleoside-modified, lipid-nanoparticle-encapsulated mRNA (mRNA-LNP) delivery in mice. We confirm specific hepatotropism of mRNA-LNP via intravenous injection of firefly luciferase encoding mRNA-LNP, with protein expression lasting about 3 days. In the liver, virtually all hepatocytes are transfected along with a subpopulation of endothelial and Kupffer cells. In homeostasis, HGF mRNA-LNP efficiently induce hepatocyte proliferation. In a chronic liver injury mouse model recapitulating non-alcoholic fatty liver disease, injections of both HGF and EGF mRNA-LNP sharply reverse steatosis and accelerate restoration of liver function. Likewise, HGF and EGF mRNA-LNP accelerate liver regeneration after acetaminophen-induced acute liver injury with rapid return to baseline ALT levels. This study introduces mRNA-LNP as a potentially translatable safe therapeutic intervention to harness liver regeneration via controlled expression of endogenous mitogens in vivo.

[1] Center for Regenerative Medicine and the Section of Gastroenterology of Boston University and Boston Medical Center, 670 Albany street, Boston, MA 02118, USA. [2] Acuitas Therapeutics, Vancouver, BC V6T 1Z3, Canada. [3] Department of Medicine, University of Pennsylvania, Philadelphia, PA 19104, USA. [4]These authors contributed equally: Fatima Rizvi, Elissa Everton. ✉email: valerige@bu.edu

Since the first evidence of the in vivo translation of naked in vitro transcribed mRNA into proteins[1], tremendous efforts have focused on modifying mRNAs to increase their stability and translatability, and decrease their inflammatory capacity[2]. Modifications include the use of modified nucleosides, removal of dsRNA contaminants, optimized 3' poly(A) tail length, integration of untranslated regions, coding sequence optimizations, and 5' cap modifications. Considerable interest in clinical use of nucleoside-modified mRNA-LNP (mRNA-LNP) has been driven by their safe, noninflammatory, nonintegrative nature, thus, eliminating any risks of mutagenesis that are associated with widespread integrative viral gene delivery systems. In addition, mRNA-LNP therapy has key advantages over recombinant protein replacement therapy including potentially improved functionality of proteins generated from mRNA-LNP given the accessibility of posttranslational cell machinery and the sustained expression of proteins, offsetting short protein half-life and allowing for intracellular delivery. Therapeutic benefits of mRNA-LNP are being investigated in clinical trials for applications ranging from cancer immunotherapy, vaccines against cancer and infectious diseases, and treatment of a variety of diseases with protein replacement therapy[3–5].

Recent liver studies have investigated the utility of mRNA-LNP in protein replacement therapy for mouse models of liver diseases that are protein deficient or express defective proteins (reviewed by Trepotec and colleagues[6]), and as a tool for in vivo gene editing of mutated fumaryl acetoacetate hydrolase using mRNA-LNP-encoded Cas9 to treat a mouse model of tyrosinemia[7]. Here, we demonstrate the application of transient, yet robust, protein production via mRNA-LNP to promptly activate regenerative pathways specifically in hepatocytes to accelerate intrinsic liver regeneration. In addition to the noninflammatory and nonintegrative properties of mRNA-LNP[8], the short-term and controllable expression of mRNA-LNP-produced proteins in the liver should prevent any deleterious consequences of long-term proliferative signals that could otherwise lead to oncogenic transformation. Therefore, mRNA-LNP constitute an ideal therapeutic strategy to activate hepatocyte regenerative pathways in the narrow time frame needed to recover liver function.

To test whether mRNA-LNP are effective tools for activating hepatocyte mitogen pathways in vivo, here we comprehensively identify the liver as the major target organ of mRNA-LNP-produced proteins following a single injection of firefly luciferase (Luc)-encoding mRNA-LNP, and show that robust protein expression lasts for about 3 days. mRNA-LNP transfect virtually all hepatocytes in the liver along with a subpopulation of endothelial cells and Kupffer cells. As a proof-of-principle of therapeutic intervention, we demonstrate that HGF mRNA-LNP efficiently induce proliferation of hepatocytes in homeostasis, and that the combination of HGF and EGF mRNA-LNP accelerates the restoration of liver function in the chronic Choline-Deficient Ethionine-supplemented (CDE) diet-induced non-alcoholic fatty liver disease (NAFLD) mouse model as well as in the acute acetaminophen-induced liver injury mouse model.

## Results

**Retro-orbital injection directs nucleoside-modified mRNA-LNP to liver.** A critical study from Pardi and colleagues compared the efficiency of mRNA-LNP to express proteins in various organs based on the route of mRNA-LNP delivery[9,10]. Intravenous (IV) injection was reported as the most effective route to efficiently target protein expression into the liver[9]. Here, we determined the organ distribution and duration of protein activity produced by firefly luciferase encoding mRNA-LNP (Luc mRNA-LNP) during a 1-week time course (at 5 h, days 1, 2, 3, 6, and 8) following a single IV injection (5 µg Luc mRNA-LNP/20 g body weight). The efficiency of two IV delivery methods including tail vein injection and retro-orbital venous sinus injection were compared. Both delivery methods led to liver restricted Luc mRNA-LNP expression as measured by bioluminescence (Fig. 1a). The highest Luc activity was detected as early

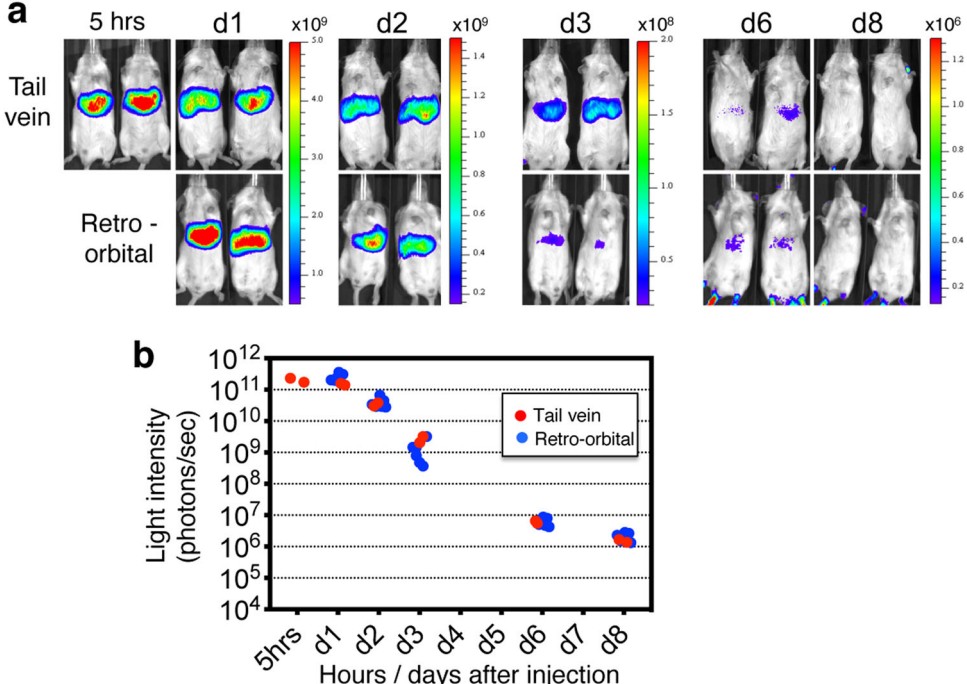

**Fig. 1 A single IV injection of Luc mRNA-LNP induces robust and restricted luciferase activity in liver. a** Images of bioluminescence measurement from 5 h, day1 (d1) up to day 8 (d8) following tail vein (n = 2 mice) or retro-orbital (n = 6 mice) injection of Luc mRNA-LNP. Scales represent light intensity in photons/sec. **b** Quantification of light intensity (photons/sec) over time. Each dot represents one mouse. Source data are provided as a Source Data file.

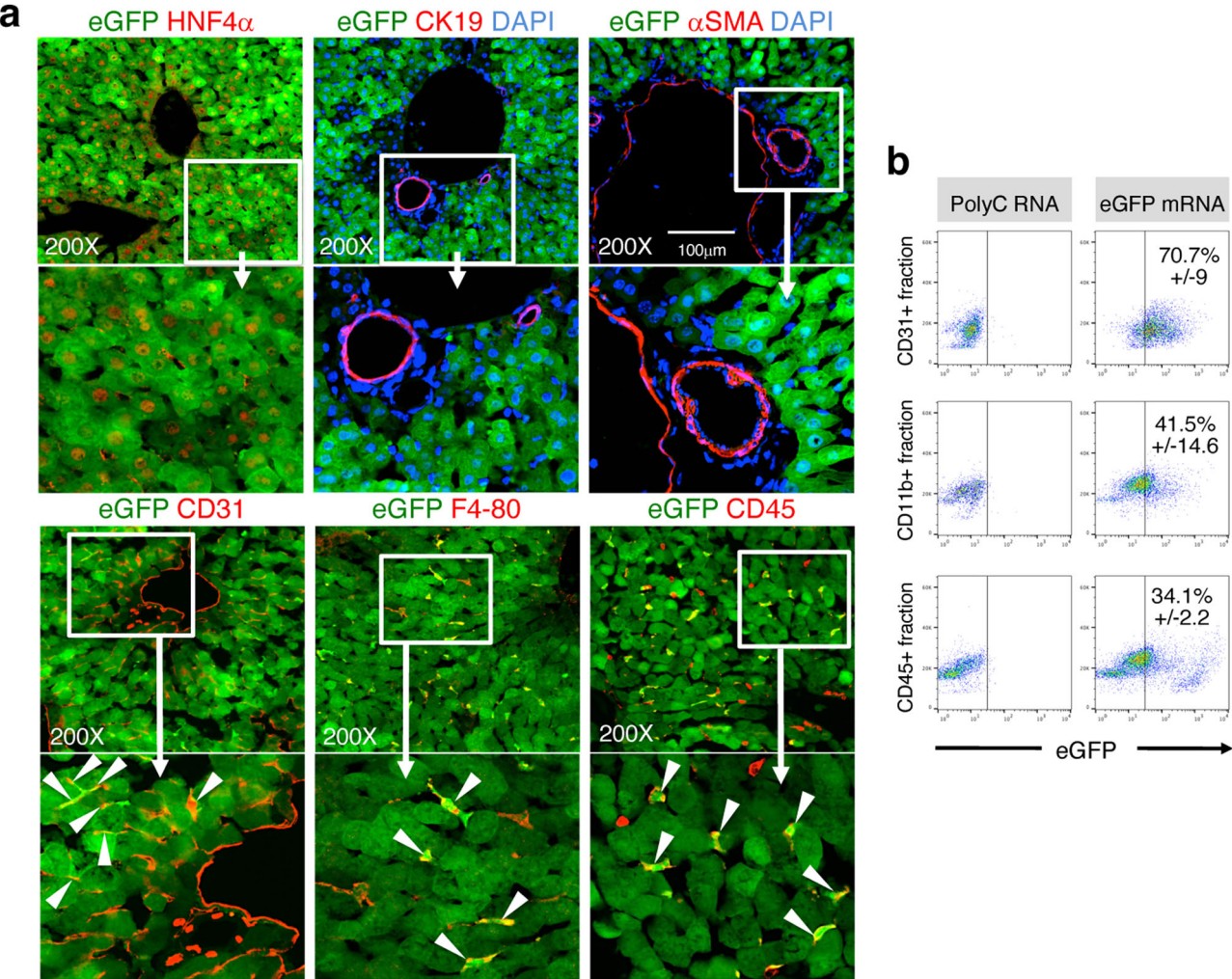

**Fig. 2 Identification of transfected liver cell types 5 h after IV eGFP mRNA-LNP injection. a** Co-immunostaining for eGFP (green) and liver cell type specific markers (red). ×200 magnification pictures and close-ups are shown. White arrowheads indicate co-stained cells. **b** Representative flow cytometry analyses showing percent of eGFP+ cells in CD11b+, CD45+, and CD31+ NPC fractions from mice injected with Poly(C) RNA-LNP or eGFP mRNA-LNP. All NPC fractions were gated for live cells present in Zombie NIR-negative populations. Data are presented as mean values ± SEM for $n = 3$ mice per group. **a** Representative images from 3 mice injected with eGFP-mRNA-LNP and 2 mice injected with Poly(C) RNA-LNP. Source data are provided as a Source Data file. The scale bar represents 100 µm for all ×200 magnification images.

as 5 h after injection, and remained stable for 24 h. Thereafter, bioluminescence decreased progressively with an average rate of one log per day to finally reach baseline activity at day 8 post-injection (Fig. 1b). Both IV delivery routes resulted in similar protein expression levels and kinetics, indicating either approach can be employed for transient expression of proteins into the liver.

**Liver-targeted nucleoside-modified mRNA-LNP predominantly transfect hepatocytes.** To further evaluate the histologic specificity of IV-injected mRNA-LNP, we performed immunostaining on sections from liver, lung, intestine, and spleen 5 h following a single injection of mRNA-LNP encoding enhanced green fluorescent protein (eGFP) (10 µg eGFP mRNA-LNP/20 g body weight) to identify eGFP+ transfected cells. While a pronounced expression was obvious in the parenchyma of the liver as early as 5 h after eGFP mRNA-LNP administration (Fig. 2a, Supplementary Fig. 1), eGFP expression was absent in parenchymal cells of the other organs analyzed (Supplementary Figs. 2, 3, 4). Notable eGFP expression was observed in the red pulp area of spleen that mainly overlapped with CD45 and CD31 expression, suggesting

that the eGFP+ cells in spleen were blood and endothelial cells (Supplementary Fig. 3). The same pattern was observed for intestine where eGFP expression was confined to the lamina propria, which contains small blood and lymph vessels as well as blood cells (Supplementary Fig. 2). In the lung, scattered eGFP+ cells were found that resembled alveolar macrophages, given their morphology, intra-alveolar luminal location, and CD45 expression (Supplementary Fig. 4). Examination of eGFP expression in various organs supports that liver is the main target organ of IV-administered eGFP mRNA-LNP, specifically the parenchymal hepatocytes, yet indicates transfection and translation in some populations of endothelial cells and hematopoietic cells in the liver, spleen, intestine, and lung as previously reported[11–13].

To identify the specific cell types transfected in the liver, we performed immunohistochemistry using characteristic markers for hepatocytes, cholangiocytes, Kupffer cells, endothelial cells, and vascular smooth muscle cells. Mice injected with LNP loaded with untranslatable Poly(C) RNA (Poly(C) RNA-LNP) were used as eGFP-negative controls. Virtually all hepatocytes identified with HNF4α expressed eGFP (Fig. 2a). eGFP expression was absent in CK19+ cholangiocytes and α-SMA+ vascular smooth muscle cells,

while few F4/80$^+$ Kupffer cells expressed eGFP. These were also represented within the CD45$^+$GFP$^+$ population of the liver. Numerous CD31$^+$ endothelial cells also exhibited notable eGFP positivity. Quantification of eGFP$^+$ cell types was further evaluated by flow cytometry following liver perfusion, separation of the nonparenchymal cell (NPC) fraction, and exclusion of dead cells. The NPC fractions yielded an average 10.5 ± 7.7% CD11b$^+$ Kupffer cells, 20.2 ± 10.5% CD45$^+$ leukocytes, and 12.1 ± 6.2% CD31$^+$ endothelial cells (Supplementary Fig. 5A, B). Significant percentages of these cell populations express eGFP, including 41.5 ± 14.6% CD11b$^+$ cells, 34.1 ± 2.2% CD45$^+$ cells, and 70.7 ± 9% CD31$^+$ cells (Fig. 2b). Altogether, immunostaining and flow cytometry analyses of eGFP mRNA-LNP- and Poly(C) RNA-LNP- injected mice indicate that hepatocytes are the main cells targeted by IV-administered mRNA-LNP, together with a significant fraction of endothelial cells and Kupffer cells. Given that hepatocyte mitogens are mostly secreted proteins, expression of mRNA-LNP encoded mitogens by endothelial and Kupffer cells would also be beneficial in activating the corresponding pathways in adjacent hepatocytes.

**Nucleoside-modified HGF mRNA-LNP induces hepatocyte proliferation in homeostasis.** HGF is known to trigger a proliferative response in hepatocytes and plays a key role during liver regeneration[14,15]. To demonstrate that the mRNA-LNP platform is an ideal delivery tool to express hepatocyte mitogens for proliferation, we examined the potential of a single IV injection of mRNA-LNP encoding the mitogen HGF to induce DNA synthesis and cell proliferation by assessing 5-ethynyl-2′-deoxyuridine (EdU) incorporation in hepatocytes in a homeostatic context (Fig. 3). When mice were given a single injection of HGF mRNA-LNP (10 μg mRNA-LNP/20 g body weight), no significant proliferative response was observed at 5 h (Fig. 3b), although HGF was strongly expressed as early as 5 h (Fig. 3a) and at later time points tested (24 and 48 h, Supplementary Fig. 6). However, over a period of 24-48 h, HGF mRNA-LNP induced a marked increase in EdU$^+$ proliferative cell numbers (Fig. 3b, e) that were mainly comprised of hepatocytes identified with HNF4α expression (Fig. 3c, white arrowheads) and very few CD45$^+$ cells (Fig. 3d, white arrowheads). Although Poly(C) RNA-LNP induced mild cell proliferation (Fig. 3b, e), EdU$^+$ cells were not identified as HNF4α$^+$ hepatocytes (Fig. 3c, blue arrowheads), but mostly as CD45$^+$ cells (Fig. 3d, white arrowheads). Quantification of the percentage of HNF4α$^+$ EdU$^+$ cells at the 48 h time point validated that the majority of EdU$^+$ cells are HNF4α$^+$ hepatocytes in HGF mRNA-LNP-treated mice (61.25 ± 10.66%), while HNF4α$^+$ EdU$^+$ cells were rarely detected in Poly(C) RNA-LNP (0.97 ± 1.66%), indicating a 121-fold increase of EdU$^+$ HNF4α$^+$ hepatocytes in HGF mRNA-LNP-treated mice compared to control Poly(C) RNA-treated mice (Fig. 3e).

**Nucleoside-modified mRNA-LNP encoding HGF and EGF accelerate restoration of liver function in a chronic liver injury model.** As a proof-of-principle for therapeutic intervention, we evaluated the ability of in vivo delivery of HGF and EGF via mRNA-LNP to rescue chronic liver injury caused by CDE diet. CDE diet has been used for decades to reproduce steatohepatitis in mice, a symptom commonly seen in human NAFLD[16]. Choline is a nutrient provided by food intake, and contributes to production of phosphatidylcholine, an essential component of cell membranes and very low-density lipoproteins, used in exporting triglycerides. Choline deficiency, therefore, causes retention of triglycerides in hepatocytes, cell dysfunction, and structural damage. Addition of ethionine, a competitive analog of the naturally-occurring amino acid methionine, further disrupts

protein synthesis in hepatocytes[17]. Characteristics of CDE diet in mice include accumulation of lipids in hepatocytes and inflammation that lead to hepatic necrosis, detected by serum liver injury markers such as alanine aminotransferase (ALT). HGF has been reported to play a role in reducing dyslipidemia associated with steatohepatitis and improving the outcomes of chronic liver injury by increasing production of antioxidant proteins and suppressing NAPDH oxidase to reduce ROS, thus, preventing hepatocyte apoptosis and subsequent liver damage[18–21]. Similarly, evidence has accumulated on the key role of epidermal growth factor receptor signaling in regenerative response of the liver upon injury through inhibition of intrahepatic lipid accumulation[22]. Given the reported synergy between HGF and EGF to promote liver regeneration by increasing cell proliferation and metabolism[15,23–26], we examined the ability of both HGF and EGF mRNA-LNP (HGF/EGF mRNA-LNP) to drive liver recovery from steatohepatitis in the CDE model. We first confirmed that HGF and EGF proteins were highly expressed in hepatocytes 24 h after a single IV injection of HGF/EGF mRNA-LNP (Supplementary Fig. 7). Injections of both HGF/EGF mRNA-LNP were performed as soon as the diet was terminated as well as 4 days later during the recovery period (Fig. 4a). Analyses of lipid accumulation in hepatocytes as assessed by Oil Red O staining on liver sections revealed a sharp decrease of steatosis 2 days after the first injection of HGF/EGF mRNA-LNP compared to that with Poly(C) RNA-LNP treatment, in which steatosis was evenly maintained with persistent macrosteatosis (Fig. 4b). At day 8 of recovery, steatosis was greatly diminished in both groups as expected with time, although the extent of steatosis remained larger in the control Poly(C) RNA-LNP-treated group. Consistent with these data, the levels of cholesterol found in the serum of mice were inversely proportional to the degree of steatosis in livers (Fig. 4c). HGF/EGF mRNA-LNP significantly accelerated the secretion of lipids from hepatocytes to the blood circulation compared to both Poly(C) RNA-LNP and Luc mRNA-LNP -treated control groups, as previously reported in rats in vivo[19] and in cultured hepatocytes in vitro[21]. Luc mRNA-loaded LNP were used as an additional negative LNP control that delivered neutral luciferase protein in hepatocytes. Single injections of either HGF or EGF mRNA-LNP decreased steatosis, although combination of both mRNA-LNP consistently resulted in significantly more efficient steatosis reversion assessed with serum cholesterol levels and Oil Red O staining (Fig. 4c, Supplementary Fig. 8). Moreover, serum levels of ALT, which are indicative of liver damage and hepatocyte necrosis, were restored to normal levels at T2, 2 days after the first injection of HGF/EGF mRNA-LNP. ALT levels in control Poly(C) RNA-LNP-treated mice were as high as T0 levels and slightly decreased in the control Luc mRNA-LNP-treated group, yet significantly greater than those in the HGF/EGF mRNA-LNP-treated group (Fig. 4d). By T8, ALT levels from all control groups dropped back to normal values as expected with time. Single injection of either HGF or EGF mRNA-LNP resulted in intermediately lower levels of ALT at T2, indicating that the combination of HGF/EGF mRNA-LNP was more efficient in restoring normal ALT levels. We confirmed that the beneficial effect of HGF/EGF mRNA-LNP was not due to influx of inflammatory cells from mRNA delivered via LNP by comparing the presence of CD45$^+$ cells in Poly(C) RNA-LNP-treated controls and HGF/EGF mRNA-LNP-treated group at T2, when the effect of HGF/EGF mRNA-LNP is the most effective at restoring liver tissue (Supplementary Fig. 9). As described in the CDE-induced liver injury model, influx of CD45$^+$ cells increased after 3 weeks of CDE diet compared to untreated mice. We noticed a similar increase in CD45$^+$ cell density in both Poly(C) RNA-LNP-treated controls and HGF/EGF mRNA-LNP-treated group, yet expectedly mild as mRNAs used in this study were

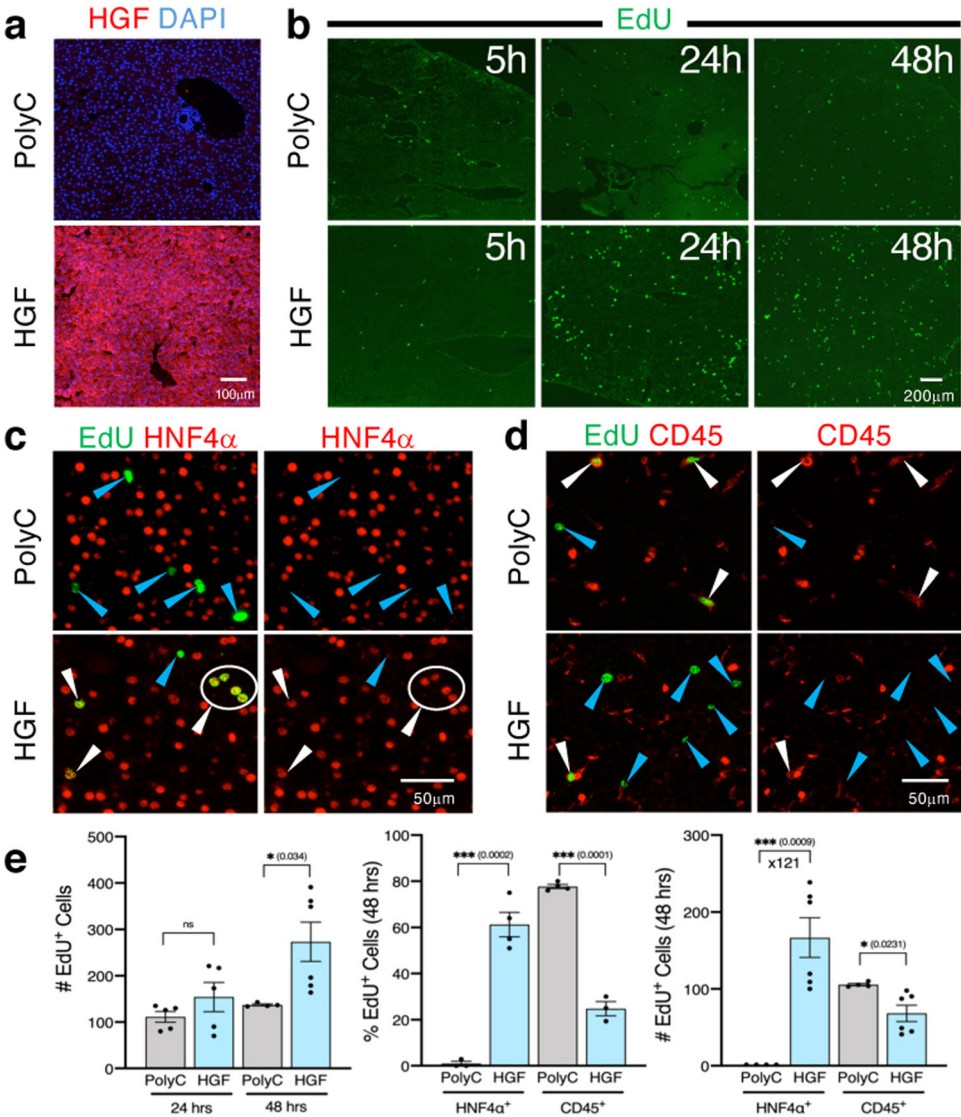

**Fig. 3 A single IV injection of HGF mRNA-LNP induces significant hepatocyte proliferation in homeostasis. a** Immunostaining for HGF in mice sacrificed 5 h after IV injection of HGF mRNA-LNP or Poly(C) RNA-LNP. Images are ×100 magnification. **b** EdU staining on liver sections of mice 5, 24, or 48 h after injection with HGF mRNA-LNP or Poly(C) RNA-LNP. ×40 magnification pictures shown. Costaining for EdU/HNF4α (**c**) or EdU/CD45 (**d**) 48 h following injection of HGF mRNA-LNP or Poly(C) RNA-LNP. White arrowheads represent costained cells, while blue arrowheads represent single EdU stained cells. Close-ups from ×200 magnification pictures are shown. **e** Quantification of the absolute numbers of EdU+ cells at ×40 magnification from 2 mice per group per time point with 2–3 fields per mouse ($n = 4$–6) following injection of either Poly(C) RNA-LNP or HGF mRNA-LNP; the percentage of EdU+ cells that are HNF4α+ (hepatocytes) or CD45+ cells (immune cells) from 1 to 2 fields at ×100 magnification from 2 mice per group at 48 h post-injection ($n = 3$–4); and the calculated absolute numbers of EdU+HNF4α+ hepatocytes and EdU+CD45+ immune cells per ×40 magnification field at 48 h based upon the counted percentages ($n = 4$–6). Data are presented as mean values ± SEM. **a–d** Representative images from 2 mice injected with either Poly(C) mRNA-LNP or HGF mRNA-LNP per time point. *P* values were calculated by two-sided student's *t* test, ***<0.0001, **<0.001, *<0.05, n.s. = not significant. Source data are provided as a Source Data file. The scale bar is included for each panel.

generated using modified nucleosides and were HPLC purified to mitigate inflammatory responses in vivo[27,28]. In addition, the majority of CD45+ cells reflected mostly the spatial pattern of CD45+ Kupffer cells within the liver, as opposed to a neutrophilic inflammatory response. These data further highlighted the specific role of HGF/EGF mRNA-LNP in restoring the liver tissue in treated mice.

To test the ability of HGF/EGF mRNA-LNP to restore liver function during persistent liver damage, mice were fed the CDE diet continuously (Fig. 5a). HGF/EGF mRNA-LNP were injected twice, once following 3 weeks of CDE diet (T0) and again 4 days later (T4) while the diet was still maintained. Mice were analyzed at T2, 2 days after the first injection and at T8. One injection of

HGF/EGF mRNA-LNP did not affect serum cholesterol, macrosteatosis as assessed by Oil Red O staining, or ALT levels at T2, compared to those in the control Poly(C) RNA-treated group (Fig. 5b, c, d). However, a second injection of HGF/EGF mRNA-LNP significantly released cholesterol into the serum, diminished steatosis, and decreased serum ALT levels at T8, suggesting that a repeated regimen of HGF/EGF mRNA-LNP injections could alleviate chronic liver damage. Of note, one Poly(C) RNA-treated mouse died while all mice treated with HGF/EGF mRNA-LNP survived, suggesting that HGF/EGF mRNA-LNP treatment promotes survival. Altogether, mRNA-LNP treatments during CDE-induced chronic injury demonstrate the clinical benefit of injections of HGF/EGF mRNA-LNP to sharply eliminate the

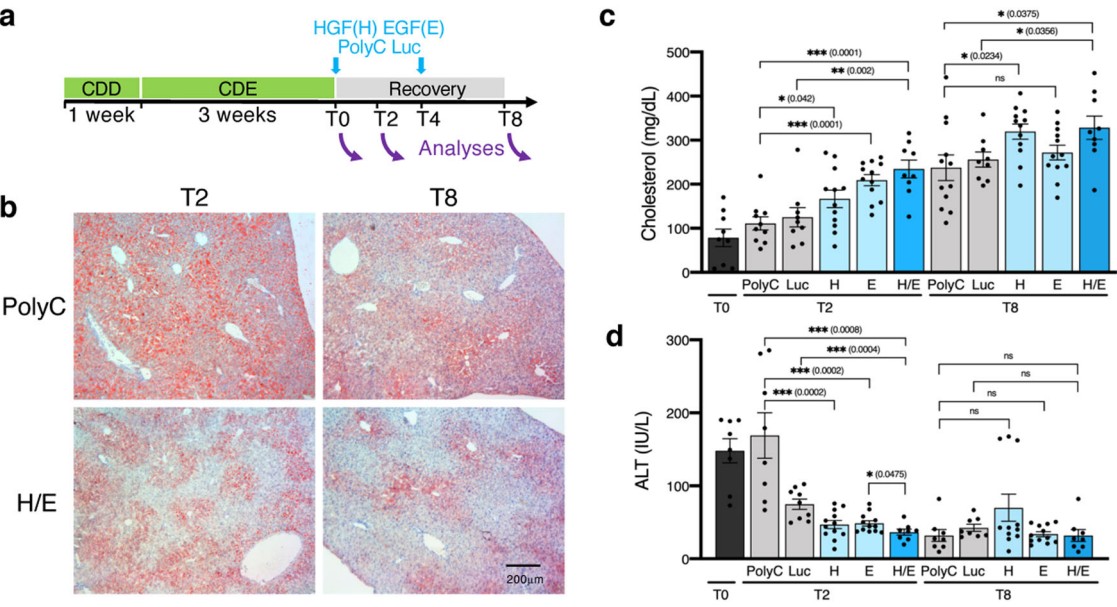

**Fig. 4 HGF/EGF mRNA-LNP accelerate restoration of liver function during recovery from CDE-induced chronic liver injury. a** Injury scheme timeline: mice were fed CDD choline deficient diet for 1 week followed by three weeks of CDD diet supplemented with 0.1% ethionine (CDE). The CDE diet was then replaced with choline-sufficient diet. Mice were injected with either controls Poly(C) RNA or Luc mRNA-LNP, or HGF(H) and EGF(E) single or combined (H/E) mRNA-LNP at the end of the CDE diet (T0), followed by another injection 4 days into recovery (T4). **b** Bright-field images at ×100 magnification of Oil Red O staining showing lipid accumulation in hepatocytes at T2 and T8 in mice injected with either Poly(C) RNA-LNP or H/E mRNA-LNP. Representative images from 3 to 4 mice per group are shown. The scale bar represents 200 μm for all ×100 magnification images. Analyses of total serum cholesterol levels (**c**) and ALT levels (**d**) from 3 to 4 mice per group in duplicate or triplicate ($n = 8$–12) per time point treated with controls Poly(C) RNA-LNP or Luc mRNA-LNP, or single HGF(H), single EGF(E) or the combination H/E mRNA-LNP indicate sharp reversion of steatosis and restoration of basic levels of ALT after H/E mRNA-LNP injections. Data are presented as mean values ± SEM. Dark grey: T0, light grey: control RNA, light blue: single mRNA, dark blue: combined mRNA. *P* values were calculated by two-sided student's *t* test, ***<0.0001, **<0.001, *<0.05, n.s. = not significant. Source data are provided as a Source Data file.

NAFLD characteristics such as lipid accumulation and liver damage.

**Nucleoside-modified mRNA-LNP encoding HGF and EGF accelerate restoration of liver function in the acute acetaminophen-induced liver injury model.** To test a broader clinical applicability of HGF/EGF mRNA-LNP to rescue liver diseases, we next evaluated the ability of HGF/EGF mRNA-LNP to accelerate liver regeneration in the well-established acetaminophen (APAP)-induced acute liver injury model[29–32] (Fig. 6a). Prior to APAP injection, female mice were subjected to the Whitten effect by exposing them to male bedding to synchronize their estrus cycle, as estrogen levels have been reported to affect APAP-induced liver damage[33]. Mice were then fasted for 14 h to allow consistent APAP-mediated liver damage among mice before injecting a single dose of APAP (550 mg/kg). A single injection of HGF/EGF mRNA-LNP was administered during the recovery period 24 h after APAP injection, and mice were analyzed 32 h and 48 h after APAP injury. Beneficial effects of mRNA-LNP were consistently and significantly observed 24 h after the single administration of HGF/EGF mRNA-LNP with accelerated disappearance of necrotic areas that were still seen in the Poly(C) RNA-LNP-treated control group assessed by H&E staining (Fig. 6b), accompanied by significantly lower serum ALT levels (Fig. 6c), and absence of TUNEL⁺ cells (Fig. 6d, e) that were still present in Poly(C) RNA-LNP control group. However, analyses at the early time point, 8 h following mRNA-LNP injection indicated no significant improvement of central vein necrotic areas (Fig. 6b), restoration of basal ALT levels (Fig. 6c), nor decrease in the extent of TUNEL⁺ apoptotic and necrotic areas (Fig. 6d, e). These results extend the clinical application of

the mitogens HGF and EGF delivered via mRNA-LNP to treat acute liver diseases.

## Discussion

Although the utility of injections of mRNA-LNP has been reported in protein replacement therapy for mouse models of liver diseases that are protein deficient or express defective proteins (reviewed by Trepotec and colleagues)[6,34], this clinical application still requires repeated injections to achieve the desired sustained expression of proteins for life. Our study introduces nucleoside-modified mRNA-LNP to a specific clinical application that requires a rapid, yet robust, protein expression to regenerate the liver. Here, we provide proof-of-principle that nucleoside-modified mRNA-LNP is an effective delivery tool to induce robust and timely controlled expression of hepatocyte mitogens and regenerative factors during the narrow time frame that is necessary to treat features of NAFLD chronic injury and to accelerate liver repair in the acute acetaminophen-induced liver injury.

Organ-specific delivery of mRNA after systemic administration remains a work-in-progress. When injected IV, mRNA-LNP mainly target the liver by binding circulating apolipoprotein E (ApoE) that in turn target ApoE receptors on the surface of hepatocytes[35]. Here, we confirm that the liver is the exclusive organ whose parenchyma is targeted, with a minor leakage in CD45⁺ blood cells, most likely monocyte populations, detected in the intestine, spleen, and lung and in a subpopulation of endothelial cells in intestine and spleen. Given the transient expression of mitogen proteins in these additional cell types and the healthy condition of the organs where they reside, we have not observed any overall health issues. However, to prevent any potential side

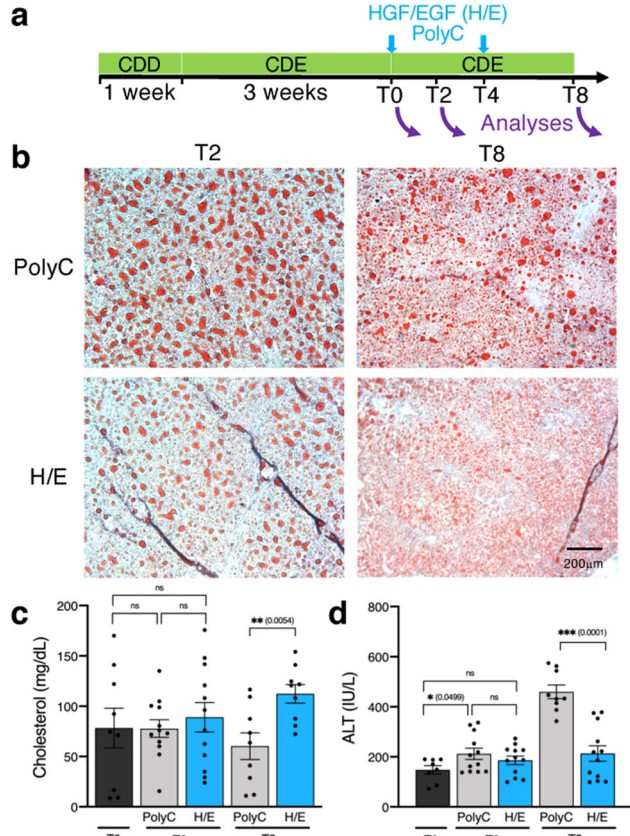

**Fig. 5 HGF/EGF mRNA-LNP ameliorate liver function during continuous CDE-induced chronic liver injury. a** Injury scheme timeline: mice were fed CDD choline deficient diet for 1 week followed by continuous CDD diet supplemented with 0.1% ethionine (CDE) for 4 weeks. Mice were injected with either Poly(C) RNA-LNP or HGF/EGF(H/E) mRNA-LNP after 3 weeks of CDE diet (T0), followed by another round of injections 4 days later (T4). Mice were analyzed at T2 and T8, 2 days and 8 days after the first mRNA-LNP administration. **b** Bright-field images at ×100 magnification of Oil Red O staining show decreased lipid content in H/E mRNA-LNP treated mice at T8 compared to T2 and compared to control Poly(C) RNA-LNP treated groups. Representative images from 4 mice per group per time point are shown. The scale bar represents 200 μm for all ×100 magnification images. Analyses of total serum cholesterol levels (**c**) and ALT levels (**d**) in 3–4 mice per group in duplicate or triplicate ($n = 8$–12) per time point indicate significant release of cholesterol into the serum and amelioration of serum ALT to baseline levels in H/E mRNA-LNP treated mice compared to those in Poly(C) RNA-LNP treated group. Data are presented as mean values ± SEM. Dark grey: T0, light grey: control RNA, blue: combined mRNA. *P* values were calculated by two-sided student's *t* test, ***<0.0001, **<0.001, *<0.05, n.s. = not significant. Source data are provided as a Source Data file.

effects in organs other than the liver, and thus to restrict expression in hepatocytes exclusively, mRNA-LNP could be conjugated with specific antibodies against hepatocytes as demonstrated for successful mRNA-LNP targeting to lung endothelial cells using an antibody against the vascular cell adhesion molecule, PECAM-1[36].

Similar to mRNA-LNP hepatotropism, asialoglycoprotein receptor (ASGPR) mediated hepatocyte targeting has been reported for gene therapy of liver diseases[37]. This strategy has also shown organ specificity limitations such as saturation of the receptor, which can reduce liver selectivity and, therefore, increase the possibility for adverse effects systemically[38]. Moreover, lower ASGPR expression observed in the event of diseases

such as hepatocellular carcinoma and hepatitis may diminish effectiveness of the ASGPR-based strategy treatment[37].

HGF has been documented to promote liver regeneration from injuries such as CCl$_4$-induced hepatocirrhosis and ischemia/reperfusion using either HGF-overexpressing mesenchymal stem cells[39–42] or HGF-encoding plasmids encased in nanoparticles[43,44]. In the more relevant clinical model of NAFLD, HGF and EGF have both been reported to alleviate steatosis, by facilitating release of lipids from hepatocytes to the blood[18,19,22,45]. Specifically, exogenous recombinant HGF proteins have shown to improve liver steatosis in a high fat diet mouse model[18], an ethanol-containing liquid diet rat model[19], and a cirrhosis combined with partial hepatectomy rat model[46]. Given the short half-life of recombinant HGF and EGF proteins, ranging from a few minutes for HGF[45,47] to less than an hour for EGF[48,49] in vivo in rodents, these experiments required daily injections[18,19] or a continuous infusion of recombinant HGF[46]. Our nucleoside-modified mRNA-LNP-based strategy overcomes the possible side effects of constitutive protein expression systems or the short clinical benefit of protein therapy. This study demonstrates robust expression of HGF and EGF in virtually all hepatocytes after a single IV injection of the corresponding nucleoside-modified mRNA-LNP. Given the sustained luciferase activity in the liver for 3 consecutive days after a single Luc mRNA-LNP injection, and robust transfection efficiency of eGFP mRNA-LNP to hepatocytes, we extrapolate that HGF and EGF activity is as robust as that of luciferase and, hence, lasts 3 days following mRNA-LNP injection. Consistent with these data, the clinical benefit of HGF/EGF mRNA-LNP treatment in regenerating the liver is demonstrated here by the sharp reversion of steatosis as well as restoration of global liver function assessed by return to baseline levels of ALT 2 days after a single injection of HGF/EGF mRNA-LNP in the chronic CDE-mediated liver injury, and acceleration of liver repair in the acute acetaminophen-induced liver toxicity. Further, the expression from these mRNAs is transient and diminishes after 4 days making it a powerful tool to achieve synthesis of desired factors in the liver over a short and controllable time frame needed to trigger liver regeneration. Overall, this study introduces the nucleoside-modified mRNA-LNP platform as an unprecedented, non-integrative, and potentially translatable safe therapeutic intervention to harness liver regeneration via timely controlled expression of mitogens or any regenerative factors in hepatocytes in vivo.

## Methods

**mRNA production.** mRNAs were produced using T7 RNA polymerase (Megascript, Ambion) on a linearized plasmid encoding codon-optimized firefly luciferase (Luc)[50,51], eGFP, epidermal growth factor (EGF), and hepatocyte growth factor (HGF). mRNAs were transcribed to contain 101 nucleotide-long poly(A) tails. One-methylpseudouridine (m1Ψ)-5'-triphosphate (TriLink) instead of UTP was used to generate modified nucleoside-containing mRNA. RNAs were capped using the m7G capping kit with 2'-O-methyltransferase (ScriptCap, CellScript) to obtain cap1. mRNA was purified by Fast Protein Liquid Chromatography (FPLC) (Akta Purifier, GE Healthcare)[52]. All mRNAs were analyzed by agarose gel electrophoresis and were stored frozen at −20°C. All nucleoside-modified mRNA are available from the company RNAx created by Dr. Drew Weissman. The Sequences of nucleoside-modified mRNA are listed in the Supplementary Table 2, Poly(C) RNAs are commercially available from Sigma.

**LNP formulation of the mRNA.** Poly(C) RNA (Sigma) and FPLC-purified m1Ψ-containing mRNAs were encapsulated in LNP using a self-assembly process in which an aqueous solution of mRNA at pH = 4.0 is rapidly mixed with a solution of lipids dissolved in ethanol[53]. LNP used in this study contain an ionizable cationic lipid (pKa in the range of 6.0–6.5, proprietary to Acuitas Therapeutics) / phosphatidylcholine / cholesterol / PEG-lipid[53,54]. The proprietary lipid and LNP composition are described in US patent US10,221,127 entitled "Lipids and lipid nanoparticle formulations for delivery of nucleic acids" (https://www.lens.org/lens/patent/183-348-727-217-109)[10]. They had a diameter of ~80 nm as measured by dynamic light scattering using a Zetasizer Nano ZS (Malvern Instruments Ltd,

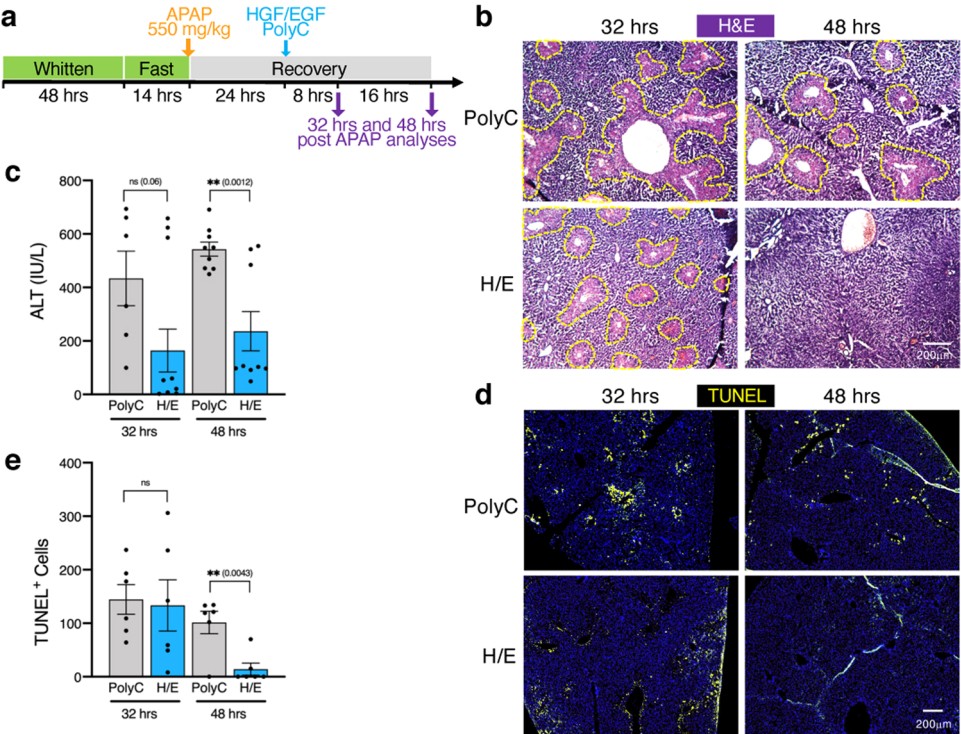

**Fig. 6 HGF/EGF mRNA-LNP accelerate liver recovery in the APAP-induced acute liver injury model. a** Mice were exposed to the Whitten effect and an overnight fast prior to a single injection of APAP (550 mg/kg). A single dose of control Poly(C) RNA-LNP or HGF/EGF(H/E) mRNA-LNP was administered 24 h following APAP injection. **b–e** Mice were analyzed 32 h and 48 h after APAP injection for necrosis with ×100 magnification bright-field images of H&E staining (**b**), serum ALT levels (**c**, 3-4 mice per group in triplicate ($n = 9$–12) per time point, except Poly(C) treated mice at 32 h for which 2 mice were used ($n = 6$)), and observation of TUNEL$^+$ necrotic and apoptotic cells in central vein liver tissue areas (**d**), that were quantified per ×40 magnification field (**e**, 3 mice per group per time point with 2 fields per mouse ($n = 6$)). **b**, **d** Representative images from 3 to 4 mice per group per time point. **c**, **e** Data are presented as mean values ± SEM. Grey: control RNA, blue: combined mRNA. *P* values were calculated by two-sided student's *t* test, ***<0.0001, **<0.001, *<0.05, n.s. = not significant. Source data are provided as a Source Data file. Scale bars are represented for each panel.

Malvern, UK) instrument. Acuitas will provide the LNP used in this work to academic investigators who would like to test it.

**Chronic and acute liver injury mouse models**. All mice used for the liver injury models are 10–12 week-old female inbred C57BL/6J from Jackson Laboratory. All animal studies were approved by the Boston University IACUC and were consistent with local, state, and federal regulations as applicable. For the chronic liver injury, mice were fed CDD choline deficient diet for 1 week followed by 3 weeks of choline-deficient diet (CDD) supplemented with 0.1% ethionine in drinking water. At the end of the regimen, CDD diet was either replaced with choline-sufficient diet along with normal drinking water during the recovery period, or continued until mice were sacrificed. Along with the liver, blood was collected from each mouse at the respective day of sacrifice. The acute liver injury was induced with a single injection of acetaminophen (APAP) 550 mg/kg (diluted in PBS) following the Whitten effect[33] and a 14-h fast, to allow synchronization of estrus cycle that may influence APAP-induced liver injury and repair and to induce consistent liver damage among the mice, respectively.

**In vivo administration of mRNA-LNP**. mRNA-LNP were thawed and freshly diluted on ice in Dulbecco's Phosphate Buffered Saline (PBS) prior to each experiment. Mice were either 10–12 week-old Bl6/C57 female for the liver injury models or 7–8 week-old Balb/c female (purchased from the Jackson Laboratory, Bar Harbor, ME) to study the kinetics of expression and identity of cell types transfected in vivo using luciferase- and eGFP mRNA-LNP at homeostasis. Mice were administered 100 µl of diluted mRNA-LNP-encoding luciferase (5 µg), eGFP (10 µg), HGF (10 µg), Poly(C) (10 µg), or combination HGF (5 µg) + EGF (5 µg) intravenously by either retro-orbital or tail vein injections using 1/2cc lo-dose insulin syringes (EXELINT).

**Bioluminescence imaging**. Bioluminescence was detected at different time-points including 5 h, 1 day, 2 days, 3 days, 6 days and 8 days after Luc-mRNA-LNP injection. Mice were anesthetized in a chamber with 2.5% isoflurane and given intra-peritoneal injections of RediJect D-Luciferin (Perkin Elmer, Waltham, MA, 150 mg/kg). Luminescence was detected with an IVIS Spectrum imaging system

(PerkinElmer, Waltham, MA) while maintaining 2% isoflurane in the imaging chamber via a nose cone. Images were captured 10 min after luciferin administration with sequence set-up of 1 s, 10 s and 20 s. The photon flux values (photons/second), corresponding to the region of interest marked around the bioluminescence signal, were analyzed using the Caliper Life Sciences software (Living IMAGE Software, Caliper).

**Tissue sources and immunohistochemistry on frozen sections**. Liver and other organs including spleen, intestine, and lung were collected directly in 4% paraformaldehyde (PFA) for 2 h-fixation at room temperature prior to OCT frozen block processing. For cryopreservation, tissues were washed thrice with PBS, dipped in 15% sucrose for 15 min, and then transferred to and kept in 30% sucrose solution until they precipitated to the bottom. The tissues were then embedded in OCT. 5 µm liver sections were cut using CM1950 Leica cryostat and slides were stored at −20°C until required for immunostaining.

The frozen sections were allowed to defrost and dry at room temperature for 30 min. The slides were dipped in PBS for 10 min and permeabilized using 0.3% triton X in PBS for 10 min. The slides were rinsed thrice in PBS, 10 min each, and blocked with 3% normal donkey serum for 30 min. The sections were then incubated overnight at 4 °C with appropriate antibody diluted in PBS at concentrations indicated in Supplementary Table 1. For EGF staining, Mouse on Mouse kit (MOM kit, Abcam) was used to reduce background from mouse tissue. Following primary antibody incubations, the slides were washed thrice with PBS, 10 min each, and incubated with the corresponding fluorescently labeled secondary antibodies for 1 h at room temperature protected from light. The slides were finally washed, incubated with DAPI for 3–5 min, rinsed, and mounted using FluorSave reagent (EMD Millipore Corp.). For the EdU staining, mice were injected intraperitoneally with 50 µg/g BW EdU in PBS 2 h prior to sacrifice. Staining was performed using Click-iT EdU Imaging Kit (C10337, Invitrogen) after the secondary antibody incubation step following manufacturers' protocol (Nikon Eclipse Ni-E microscope).

**Liver dissociation and flow cytometry**. Liver was perfused according to the previously published method with minor modifications[55]. All solutions were

pre-warmed to 40 °C and delivered at a rate of 4 ml/min. Mice were anaesthetized with ketamine/xylazine and then perfused by cannulation with a 24-gauge catheter through the inferior vena cava with 30 mL 1X Liver Perfusion Medium (Gibco by Life Technologies), while the portal vein was cut to allow the perfusate to flow out. In total, 15 mL of Earle's Balanced Salt Solution (EBSS) containing 10 mM Hepes (Ca++ and Mg++, pH 7.4) was then perfused, and finally followed with 30 mL of Liver Digest Medium (Gibco by Life Technologies) to allow complete dissociation of liver cells in situ. Livers were then extracted and mechanically dissociated in 10 mL of Liver Digest Medium. Cells were filtered through 100 μm cell strainer and filtrate was centrifuged at $50 \times g$ for 2 min at 4 °C to obtain hepatocytes in the pellet while the supernatant was collected as the fraction of NPCs. The hepatocyte pellet was resuspended in wash media (Hepatocyte Wash Medium by Gibco + 0.1 mg/mL DNAse 1 + 10% FBS). The cell fraction caught in the 100 μm strainer was further digested in NPC Digest Medium (2.5 mg/mL Collagenase IV + 0.1 mg/mL DNAse 1), filtered through a 40 μm strainer, then centrifuged at $300 \times g$ for 5 min at 4 °C. The supernatant was discarded, pellet was resuspended in wash media, and solution was added to the other collection of NPCs. Following 10-min incubations in wash media, both fractions were washed then resuspended in PBS. Fractions were incubated in Zombie NIR Fixable Viability dye (1:2000, Biolegend # 423105) for 30 min at room temperature, washed, and resuspended in FACS buffer (2% FBS + 2 mM EDTA at pH 8). Fractions were then separated into 100 μl wells and treated with 1 μg/100 μl Fc Block (BD Pharmingen #553141) for 10 min at room temperature. Wells were then incubated with 0.5 μg CD45-PEeFluor 610, 0.25 μg CD11b-PEeFluor 610, or 0.5 μg CD31-PE/Dazzle 594 conjugated antibodies, or their isotypes Rat-PEeFluor 610 (Invitrogen 61-4031-80) and Rat-PE/Dazzle 594 (Biolegend #400557) for 20 min at room temperature. Following antibody incubation, cells were washed and resuspended in FACS buffer. Cells were run on a Stratedigm S1000EXi flow cytometer and data analyzed with FlowJo v10 software. Care was taken to include proper unstained and isotype controls to derive the compensation matrix.

**ALT assay.** Assays were performed using the Pointe Scientific kit (A7526-450) for testing serum ALT levels following manufacturers protocol. Briefly, 10 μl of serum was mixed with supplied reagent mix at 37 °C and readings were measured at 340 nm every 1 min for 5 min using Molecular Devices SpectraMax® i3x Multi-Mode microplate reader.

**Oil red O assay.** Lipid staining was performed on frozen-fixed liver tissue sections using iso-propanol method. Briefly, sections were rinsed with 60% iso-propanol and stained with freshly prepared Oil Red O solution for 15 min. Slides were rinsed twice with 60% iso-propanol and nuclei were lightly stained using hematoxylin solution. Slides were washed, mounted, and observed under bright-field microscope.

**Total cholesterol assay.** Assays were performed using the Total Cholesterol E Assay from FUJIFILM Wako Diagnostics (#999-02601) using manufacturers protocol. We adapted the protocol for a 96-well plate by using 5 μL of serum with 200 μL of test reagent. After incubation at 37 °C for 5 min, reactions were read at 600 nm using Molecular Devices SpectraMax® i3x Multi-Mode microplate reader and cholesterol levels calculated from a standard curve.

**Hematoxylin & eosin (H&E) assay.** Histology was performed on frozen-fixed liver tissue sections. Briefly, slides were hydrated with tap water, stained with Gill's Hematoxylin, blued with ammonia, washed with ethanol, stained with 0.25% Eosin Y, then cleared with Histoclear. Slides were mounted with permanent mounting media and observed under bright-field microscope.

**TUNEL assay.** Apoptotic and necrotic cells were visualized and quantified using the Invitrogen Click-iT™ Plus TUNEL Assay for In Situ Apoptosis Detection (Alexa Fluor™ 488 dye). The protocol was adapted for use on frozen-fixed tissue sections by scaling up reagent proportions for larger volumes. Sections were counter-stained with DAPI, mounted, and observed under a fluorescence microscope green channel (Nikon Eclipse Ni-E microscope).

**Statistics and reproducibility.** The statistical analyses were carried out using two-sided student's $t$ test for unpaired comparisons with GraphPadPrism. A $p$ value < 0.05 was considered significant, ***<0.0001, **<0.001, *<0.05, n.s. = not significant. The results are presented as the means ± standard error or standard deviation as indicated in legends.

**Reporting summary.** Further information on research design is available in the Nature Research Reporting Summary linked to this article.

## Data availability

All data generated or analyzed during this study are included in this published article and its supplementary information. Source data are provided with this paper.

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

## Acknowledgements
This work was supported by the March of Dimes Research Grant #6-FY14-530, NIH NIDDK R01DK124361-01A1, and the Alpha-1 Foundation Research Grant ID614163. We are grateful to Brian R. Tilton of the BUSM Flow Cytometry Core for technical assistance, supported by NIH Grant 1UL1TR001430, and Drs. Greg Miller and Marianne James of the CReM, supported by grants R24HL123828 and U01TR001810. Dr. Hua Liu, Elissa Everton and Anna Rose Smith are supported by the TL1TR001410 award.

## Author contributions
F.R., E.E., A.R.S., H.L., N.P., D.W., and V.G.-E. conceived and designed experiments. F.R., E.E., A.R.S., H.L., and E.O. performed experiments. M.B. and Y.T. designed and prepared the L.N.P. F.R., E.E., A.R.S., H.L., N.P., and V.G.-E. wrote the manuscript. V.G.-E. directed the research.

## Competing interests
In accordance with the University of Pennsylvania policies and procedures and our ethical obligations as researchers, we report that Drew Weissman is named on patents that describe the use of nucleoside-modified mRNA as a platform to deliver therapeutic proteins. Relevant to this study, Drew Weissman and Norbert Pardi are also named on a patent describing the use of modified mRNA in lipid nanoparticles US patent US8,278,036 entitled "RNA containing modified nucleosides and methods of use thereof". Mitchell Beattie and Ying Tam are employees of Acuitas Therapeutics, a company focused on the development of lipid nanoparticulate nucleic acid delivery systems for therapeutic applications. All other authors declare no competing interests.
