## [Peer Review File · Nature Communications]

Reviewers' Comments:

Reviewer #1:

Remarks to the Author:

Rizvi et al. examine the expression of mRNAs in the liver via LNP delivery. The manuscript is concise and well written. For the most part, the experiments are well performed, though there are a few controls that should have been included instead of polyC RNA. I have the following concerns:

- 1) Figure 3 C/D. These panels are poorly labeled. "EdU HNF4" does not tell me what is different about the left panel vs. the right one. Likewise for panel D.
- 2) Figure 3. A graph indicating the values of dividing cells should be included as a new panel E (and not just in the text).
- 3) Figure 3 and 4. Why was polyC mRNA used as the control instead of Luciferase or GFP? polyC has no 5' cap, no ORF, no poly A tail and is not made by T7 polymerase. Having the correct control here is of significance, especially for Figure 4.
- 4) Figure 4. Why are there no panels of singular HGF and EGF?
- 5) I understand the need for secrecy regarding Acuitas' lipid, but if the patent is published (as listed in the methods), the lipid used should be identified as well as some aspects of its biophysical properties.
- 6) There are no controls of immuno-stimulation and cytokine release. This is critical for understanding Figure 4.

Reviewer #2:

Remarks to the Author:

This manuscript reports on a system to safely and transiently express HGF and EGF mRNAs specifically to the liver for the treatment of serious liver diseases, in this case non-alcoholic fatty liver disease (NAFLD), a disease of almost epidemic proportions in the Western world. The manuscript is a little limited in that only one liver disease (but important one) is explored, simply showing fatty infiltration is reduced more rapidly with the treatment along with a quicker reduction in liver enzymes that signal hepatocyte injury, along with a more rapid rise in blood cholesterol. Thus, it would have been good to show how the treatments affected other diseases, particularly models of acute liver injury (galactosamine or repeated carbon tetrachloride) and 'small for size' syndromes that plague cases where liver large scale partial hepatectomy is required for tumour resection. This is particularly pertinent since the title states '... activation of regenerative pathways..' - cell regenerative pathways have not so far been explored save the hyperplasia induced in the normal liver. The paper is very well written and illustrated apart from several errors in labeling.

Minor points:

1. Suppl Figs 6 and 7: legends state 'protein', the abscissas state 'mRNA'
2. No details of EdU labelling, dose? Was this a pulse chase or were injections given immediately prior to killing?

Reviewer #3:

Remarks to the Author:

The manuscript "Liver repair via transient activation of regenerative pathways in hepatocytes using lipid nanoparticle-complexed nucleoside-modified mRNA" by Rizvi et al described a method to transiently express hepatocyte growth factor and EGF using lipid nanoparticles. The authors use this nanoparticle platform to deliver modified mRNA encoding EGF and HGF to the liver to induce hepatocyte proliferation. They then examine this in a liver injury model and show improvement in liver function. The authors use a lipid nanoparticle that is not novel as has been widely leveraged in the community. Unsurprisingly in the mouse the main organ that this was distributed to was the liver but also uptake was seen in many other organ macrophages (lung/spleen). Furthermore the use of HGF systemically for improvement of liver regeneration/repair has been evaluated previously both systemically, using cells, and using nanoparticles (PMID 22963044, 24634942, 31058711, 32245503). Overall the findings will be greatly improved with greater discussion of the findings, improved placement of its findings with prior reports therefore the manuscript as it currently stands along with the comments below means that the manuscript would need to be significantly revised.

Major Comments for the author:

1. The authors do not examine the impact of overexpression in other organs by the macrophage populations of HGF/EGF. Potential adverse effects on both this cellular population as well as the organ itself should be evaluated and considered. A more targeted approach to the hepatocyte to exclude or decrease this possibility such as using ASGPR targeting (galactosylation) or other approaches should have been considered and discussed.
2. The authors demonstrate proliferation with EDU staining. However they also define the cell that is proliferating with a single nuclear marker. Although HNF4a is a good hepatocyte marker an additional marker such as albumin or other cytoplasmic markers would lend greater confidence in these imaging findings.
3. The authors should quantify the number of proliferating cells for each population in the liver (e.g. CD45+/EDU+; HNF4a/EDU+ etc).
4. The CDE model is a popular model to study mouse liver injury but does not readily phenocopy human liver disease or metabolic disease. The groups findings should be validated in an additional model of liver disease demonstrating that the group's technology and findings are not due to an artifact of this model. The groups main findings are that it sped up repair but not that it changed the endpoint or outcome in these experiments. The repair would have occurred regardless of the group's intervention. Dosing in the context of active injury would greatly show the impact of the intervention developed by the authors.

Point-by-point response to the reviewers' comments, reproduced verbatim

We greatly appreciated the comments from the three reviewers in helping us improve the quality of the manuscript.

Reviewer #1

Rizvi et al. examine the expression of mRNAs in the liver via LNP delivery. The manuscript is concise and well written. For the most part, the experiments are well performed, though there are a few controls that should have been included instead of polyC RNA. I have the following concerns:

1) Figure 3 C/D. These panels are poorly labeled. "EdU HNF4" does not tell me what is different about the left panel vs. the right one. Likewise for panel D.

Response: We apologize for the confusing labels. Those have been fixed.

2) Figure 3. A graph indicated the values of dividing cells should be included as a new panel E (and not just in the text).

Response: This is a good point. We have included a new panel E that illustrates the numbers of EdU⁺ cells (first graph), percentages (second graph) and absolute numbers (third graph) of EdU⁺HNF4 α ⁺ hepatocytes and EdU⁺CD45⁺ blood cells. In the text (page 5, second paragraph), we included: "Quantification of the percentage of HNF4 α ⁺ EdU⁺ cells at the 48 hour time point validated that the majority of EdU⁺ cells are HNF4 α ⁺ hepatocytes in HGF mRNA-LNP-treated mice (61.25 +/- 10.66%), while HNF4 α ⁺ EdU⁺ cells were rarely detected in PolyC RNA-LNP (0.97 +/- 1.66%), indicating a 121-fold increase of EdU⁺HNF4 α ⁺ hepatocytes in HGF mRNA-LNP-treated mice compared to control PolyC RNA-treated mice (Fig. 3E)."

3) Figure 3 and 4. Why was polyC mRNA used as the control instead of Luciferase or GFP? polyC has no 5' cap, no ORF, no poly A tail and is not made by T7 polymerase. Having the correct control here is of significance, especially for Figure 4.

Response: We addressed this important point in Figure 4, as recommended. In addition to the PolyC loaded LNP, we have utilized LNP loaded with Luciferase mRNA, and showed that the combination of HGF/EGF mRNA-LNP greatly reduce steatosis by inducing release of cholesterol in serum and decrease of Oil Red O staining as well as restoring baseline levels of serum ALT compared to both PolyC RNA-LNPs and Luciferase mRNA-LNP with the same significance ($p < 0.005$) for both controls (new Figure 4C and 4D, new Supplementary Figure 8). Please see text page 6, first paragraph.

4) Figure 4. Why are there no panels of singular HGF and EGF?

Response: This is a very fair point. Our preliminary data indicated that the combination of both factors was more efficient than single factors. These results brought us to only include the data from the combined factors in the first submission. We have now completed our preliminary data and statistically compared the effect of single factors with the combined factors. We showed that single injections of either HGF or EGF mRNA-LNPs decreased steatosis, although combination of both mRNA-LNP consistently resulted in significantly more efficient steatosis reversion (Fig. 4C, Supplementary Fig. 8). Likewise, single injection of either HGF or EGF mRNA-LNP resulted

in intermediately lower levels of ALT at T2, indicating that the combination of HGF/EGF mRNA-LNP was more efficient in restoring normal ALT levels (Fig. 4D). Please see text page 6, first paragraph.

5) *I understand the need for secrecy regarding Acuitas' lipid, but if the patent is published (as listed in the methods), the lipid used should be identified as well as some aspects of its biophysical properties.*

Response: Please see below the edited version of the method related to the LNP with the new addition in red: "LNP formulation of the mRNA: PolyC-RNA (Sigma) and FPLC-purified m1Ψ-containing mRNAs were encapsulated in LNP using a self-assembly process in which an aqueous solution of mRNA at pH=4.0 is rapidly mixed with a solution of lipids dissolved in ethanol³³. LNPs used in this study were similar in composition to those described previously^{33,34}, which contain an ionizable cationic lipid (pKa in the range of 6.0-6.5, proprietary to Acuitas Therapeutics) / phosphatidylcholine / cholesterol / PEG-lipid. The proprietary lipid and LNP composition are described in US patent US10,221,127. They had a diameter of ~80 nm as measured by dynamic light scattering using a Zetasizer Nano ZS (Malvern Instruments Ltd, Malvern, UK) instrument. **Acuitas will provide the LNP used in this work to academic investigators who would like to test it.**"

6) *There are no controls of immuno-stimulation and cytokine release. This is critical for understanding Figure 4.*

Response: We have addressed this comment by showing that the abundance of CD45⁺ cells that include Kupffer cells of the liver and any immune cells from the blood, was similar in control Poly(C) RNA-LNP and EGF/HGF mRNA-LNP treated groups. This is commented in page 6 at the end of the first paragraph as follows:

"We confirmed that the beneficial effect of HGF/EGF mRNA-LNP was not due to influx of inflammatory cells from mRNA delivered via LNP by comparing the presence of CD45⁺ in Poly(C) RNA-LNP-treated controls and HGF/EGF mRNA-LNP-treated group at T2, when the effect of HGF/EGF mRNA-LNP is the most effective at restoring liver tissue (Supplementary Fig. 9). As described in the CDE-induced liver injury model, influx of CD45⁺ cells increased after 3 weeks of CDE diet compared to untreated mice. We noticed a similar increase in CD45⁺ cell density in both Poly(C) RNA-LNP-treated controls and HGF/EGF mRNA-LNP-treated group, yet expectedly mild as mRNAs used in this study were generated using modified nucleosides and were HPLC purified to mitigate inflammatory responses in vivo^{26,27}. Additionally, the majority of CD45⁺ cells reflected mostly the spatial pattern of CD45⁺ Kupffer cells within the liver, as opposed to a neutrophilic inflammatory response. These data highlighted further the specific role of HGF/EGF mRNA-LNP in restoring the liver tissue in treated mice."

Reviewer #2

This manuscript reports on a system to safely and transiently express HGF and EGF mRNAs specifically to the liver for the treatment of serious liver diseases, in this case non-alcoholic fatty liver disease (NAFLD), a disease of almost epidemic proportions in the Western world. The manuscript is a little limited in that only one liver disease (but important one) is explored, simply showing fatty infiltration is reduced more rapidly with the treatment along with a quicker reduction in liver enzymes that signal hepatocyte injury, along with a more rapid rise in blood cholesterol. Thus, it would have been good to show how the treatments affected other diseases, particularly models of acute liver injury (galactosamine or repeated carbon tetrachloride) and

'small for size' syndromes that plague cases where liver large scale partial hepatectomy is required for tumour resection. This is particularly pertinent since the title states '... activation of regenerative pathways..'- cell regenerative pathways have not so far been explored save the hyperplasia induced in the normal liver. The paper is very well written and illustrated apart from several errors in labeling.

Response: We thank the reviewer for this key comment. We have now included an additional liver injury model which is an acute injury, as recommended, induced with a single dose of acetaminophen (APAP). Data from this new model constitute the new Figure 6. We showed that mRNA-LNP encoding HGF and EGF accelerate restoration of liver function following APAP injury assessed with disappearance of necrotic areas and absence of TUNEL⁺ cells, accompanied by significantly lower serum ALT levels. Please see text page 7, first paragraph.

Minor

points:

1. *Suppl Figs 6 and 7: legends state 'protein', the abscissas state 'mRNA'*
2. *No details of EdU labelling, dose? Was this a pulse chase or were injections given immediately prior to killing?*

Response: Thank you for catching these points. *Suppl Figs 6 and 7:* To be consistent throughout the manuscript, we have simplified the abscissas by writing only PolyC and HGF or EGF when the corresponding mRNA/RNA-LNP were injected. The legends remain the same. *EdU labelling dose* is now indicated in the Materials and Methods section under the section “Tissue sources and immunohistochemistry on frozen sections”. Mice were injected intraperitoneally with 50µg/g BW EdU in PBS two hours prior to sacrifice.

Reviewer #3

The manuscript “Liver repair via transient activation of regenerative pathways in hepatocytes using lipid nanoparticle-complexed nucleoside-modified mRNA” by Rizvi et al described a method to transiently express hepatocyte growth factor and EGF using lipid nanoparticles. The authors use this nanoparticle platform to deliver modified mRNA encoding EGF and HGF to the liver to induce hepatocyte proliferation. They then examine this in a liver injury model and show improvement in liver function. The authors use a lipid nanoparticle that is not novel as has been widely leveraged in the community. Unsurprisingly in the mouse the main organ that this was distributed to was the liver but also uptake was seen in many other organ macrophages (lung/spleen). Furthermore the use of HGF systemically for improvement of liver regeneration/repair has been evaluated previously both systemically, using cells, and using nanoparticles (PMID 22963044, 24634942, 31058711, 32245503). Overall the findings will be greatly improved with greater discussion of the findings, improved placement of its findings with prior reports therefore the manuscript as it currently stands along with the comments below means that the manuscript would need to be significantly revised.

Response about the “lipid nanoparticle that is not novel as has been widely leveraged in the community”: We have indeed discussed this point in the discussion section in the second paragraph of page 7 as follows: “Although the utility of injections of mRNA-LNP has been reported in protein replacement therapy for mouse models of liver diseases that are protein deficient or express defective proteins (reviewed by Trepotec and colleagues⁶ and ³³), this clinical application still requires repeated injections to achieve the desired sustained expression of proteins for life. Our study introduces nucleoside-modified mRNA-LNP to a specific clinical application that

requires a rapid, yet robust, protein expression to regenerate the liver. Here, we provide proof-of-principle that nucleoside-modified mRNA-LNP is an effective delivery tool to induce robust and timely controlled expression of hepatocyte mitogens and regenerative factors during the narrow time frame that is necessary to treat features of NAFLD chronic injury and to accelerate liver repair in the acute acetaminophen-induced liver injury.

Response about *“the use of HGF systemically for improvement of liver regeneration/repair has been evaluated previously both systemically, using cells, and using nanoparticles (PMID 22963044, 24634942, 31058711, 32245503)”*: These references among others have been included in the discussion section in the second paragraph of page 8 and the benefit of our mRNA-LNP compared to those studies was discussed as follows: “ HGF has been documented to promote liver regeneration from injuries such as CCl₄-induced hepatocirrhosis and ischemia/reperfusion using either HGF-overexpressing mesenchymal stem cells³⁸⁻⁴¹ or HGF-encoding plasmids encased in nanoparticles^{42,43}. In the more relevant clinical model of NAFLD, HGF and EGF have both been reported to alleviate steatosis, by facilitating release of lipids from hepatocytes to the blood^{17, 18, 21, 44}. Specifically, exogenous recombinant HGF proteins have shown to improve liver steatosis in a high fat diet mouse model¹⁷, an ethanol-containing liquid diet rat model¹⁸, and a cirrhosis combined with partial hepatectomy rat model⁴⁵. Given the short half-life of recombinant HGF and EGF proteins, ranging from a few minutes for HGF⁴⁴⁻⁴⁶ to less than an hour for EGF^{47,48} *in vivo* in rodents, these experiments required daily injections^{17,18} or a continuous infusion of recombinant HGF⁴⁵. Our nucleoside-modified mRNA-LNP-based strategy overcomes the possible side effects of constitutive protein expression systems or the short clinical benefit of protein therapy. This study demonstrates robust expression of HGF and EGF in virtually all hepatocytes after a single IV injection of the corresponding nucleoside-modified mRNA-LNP. Given the sustained luciferase activity in the liver for 3 consecutive days after a single Luc mRNA-LNP injection, and robust transfection efficiency of eGFP mRNA-LNP to hepatocytes, we extrapolate that HGF and EGF activity is as robust as that of luciferase and, hence, lasts 3 days following mRNA-LNP injection...”

Major Comments for the author:

1. *The authors do not examine the impact of overexpression in other organs by the macrophage populations of HGF/EGF. Potential adverse effects on both this cellular population as well as the organ itself should be evaluated and considered. A more targeted approach to the hepatocyte to exclude or decrease this possibility such as using ASGPR targeting (galactosylation) or other approaches should have been considered and discussed.*

Response: Thank you for this comment. We have discussed this issue in the discussion section in the third paragraph of page 7 as follows: “Organ-specific delivery of mRNA after systemic administration remains a work-in-progress. When injected IV, mRNA-LNP mainly target the liver by binding circulating apolipoprotein E (ApoE) that in turn target ApoE receptors on the surface of hepatocytes³⁴. Here, we confirm that the liver is the exclusive organ whose parenchyma is targeted, with a minor leakage in CD45⁺ blood cells most likely monocyte populations, detected in the intestine, spleen and lung and in a subpopulation of endothelial cells in intestine and spleen. Given the transient expression of mitogen proteins in these additional cell types and the healthy condition of the organs where they reside, we have not observed any overall health issues. However, to prevent any potential side effects in organs other than the liver, and thus to restrict expression in hepatocytes exclusively, mRNA-LNP could be conjugated with specific antibodies against hepatocytes as demonstrated for successful mRNA-LNP targeting to lung endothelial cells using an antibody against the vascular cell adhesion molecule, PECAM-1³⁵.”

We thank the reviewer for bringing our attention to the asialoglycoprotein receptor (ASGPR) mediated hepatocyte targeting strategy for gene therapy of liver diseases. We have now included in the discussion a more thorough discussion of this strategy as follows in the fourth paragraph of page 7: “Similar to mRNA-LNP hepatotropism, asialoglycoprotein receptor (ASGPR) mediated hepatocyte targeting has been reported for gene therapy of liver diseases³⁶. This strategy has also shown organ specificity limitations such as saturation of the receptor, which can reduce liver selectivity and, therefore, increase the possibility for adverse effects systemically³⁷. Moreover, lower ASGPR expression observed in the event of diseases such as hepatocellular carcinoma and hepatitis may diminish effectiveness of the ASGPR-based strategy treatment³⁶.”

2. The authors demonstrate proliferation with EDU staining. However they also define the cell that is proliferating with a single nuclear marker. Although HNF4a is a good hepatocyte marker an additional marker such as albumin or other cytoplasmic markers would lend greater confidence in these imaging findings.

Response: As recommended, we have performed albumin immunostaining on liver sections. However, given the patchy albumin staining among hepatocytes, as shown below from 3 different fields, albumin staining didn't reveal as a reliable marker to consistently identify all hepatocytes. We believe that the nuclear marker HNF4 α together with the polygonal cytoplasmic shape of hepatocytes provided a faithful identity of hepatocytes in this study.

3. The authors should quantify the number of proliferating cells for each population in the liver (e.g. CD45⁺/EDU⁺; HNF4a/EDU⁺ etc).

Response: This is an important point that we carefully addressed and included in the new panel E of Figure 3.

4. The CDE model is a popular model to study mouse liver injury but does not readily phenocopy human liver disease or metabolic disease. The groups findings should be validated in an additional model of liver disease demonstrating that the group's technology and findings are not due to an artifact of this model. The groups main findings are that it sped up repair but not that it changed the endpoint or outcome in these experiments. The repair would have occurred regardless of the group's intervention. Dosing in the context of active injury would greatly show the impact of the intervention developed by the authors.

Response: These two points are critical and have been addressed by including two new figures: Figure 5 and 6.

Figure 5 tests the ability of HGF/EGF mRNA-LNP to restore liver function during persistent liver damage when mice are fed the CDE diet continuously. We showed that two injections of

HGF/EGF mRNA-LNP significantly released cholesterol into the serum, diminished steatosis, and decreased serum ALT levels at T8, suggesting that a repeated regimen of HGF/EGF mRNA-LNP injections could alleviate chronic liver damage.

Figure 6 demonstrates that HGF/EGF mRNA-LNP accelerate restoration of liver function in the acute acetaminophen-induced liver injury model.

Reviewers' Comments:

Reviewer #1:

Remarks to the Author:

The authors have greatly improved the study.

Reviewer #2:

Remarks to the Author:

The authors appear to have responded positively to the comments of the 3 referees, in particular incorporating an additional model of liver injury that supports their original conclusions.

Reviewer #3:

Remarks to the Author:

The manuscript "Liver repair via transient activation of regenerative pathways in hepatocytes using lipid nanoparticle-complexed nucleoside-modified mRNA" by Rizvi et al is a resubmission of a prior manuscript which describes a method to transiently express hepatocyte growth factor and EGF using lipid nanoparticles. The authors use this nanoparticle platform to deliver modified mRNA encoding EGF and HGF to the liver to induce hepatocyte proliferation. They then examine this in two liver injury model and show improvement in liver function. The authors agree that the use of a lipid nanoparticle in itself is not novel and address this in their introduction and discussion. Moreover they also address the concern that the use of use of HGF systemically for improvement of liver regeneration/repair has also been used in several studies. Additions in the discussion to put this paper into better context of the filed greatly strengthens the manuscript. Several additions were made to address this reviewer's concerns including quantification of proliferating cells for the populations in the liver. While addition of an additional marker for the hepatocytes would have strengthened the author's claims (e.g. albumin was a suggestion not a hard target (as the authors could have tried other markers such as KRT 18 etc); the addition of the additional injury model and overall text and figure additions greatly strengthen the manuscript. In this substantially revised manuscript the manuscript should be accepted.

Point-by-point response to the reviewers' comments

Based on the comments from the 3 reviewers copied below, there was no additional modification to include in the manuscript.

Reviewer #1 (Remarks to the Author):

The authors have greatly improved the study.

Reviewer #2 (Remarks to the Author):

The authors appear to have responded positively to the comments of the 3 referees, in particular incorporating an additional model of liver injury that supports their original conclusions.

Reviewer #3 (Remarks to the Author):

The manuscript "Liver repair via transient activation of regenerative pathways in hepatocytes using lipid nanoparticle-complexed nucleoside-modified mRNA" by Rizvi et al is a resubmission of a prior manuscript which describes a method to transiently express hepatocyte growth factor and EGF using lipid nanoparticles. The authors use this nanoparticle platform to deliver modified mRNA encoding EGF and HGF to the liver to induce hepatocyte proliferation. They then examine this in two liver injury model and show improvement in liver function. The authors agree that the use of a lipid nanoparticle in itself is not novel and address this in their introduction and discussion. Moreover they also address the concern that the use of HGF systemically for improvement of liver regeneration/repair has also been used in several studies. Additions in the discussion to put this paper into better context of the filed greatly strengthens the manuscript. Several additions were made to address this reviewer's concerns including quantification of proliferating cells for the populations in the liver. While addition of an additional marker for the hepatocytes would have strengthened the author's claims (e.g. albumin was a suggestion not a hard target (as the authors could have tried other markers such as KRT 18 etc); the addition of the additional injury model and overall text and figure additions greatly strengthen the manuscript. In this substantially revised manuscript the manuscript should be accepted.